# The Association of Work Satisfaction and Burnout Risk in Endoscopy Nursing Staff—A Cross-Sectional Study Using Canonical Correlation Analysis

**DOI:** 10.3390/ijerph17082964

**Published:** 2020-04-24

**Authors:** Charles Christian Adarkwah, Oliver Hirsch

**Affiliations:** 1Department of General Practice and Family Medicine, Philipps-University, 35043 Marburg, Germany; charles.adarkwah@uni-siegen.de; 2CAPHRI School for Public Health and Primary Care, Department of Health Services Research, Maastricht University, 6200 Maastricht, The Netherlands; 3Faculty of Life Sciences, University of Siegen, 57076 Siegen, Germany; 4FOM University of Applied Sciences, 57078 Siegen, Germany

**Keywords:** work satisfaction, burnout, endoscopy, correlation of data, factor analysis

## Abstract

Background: Burnout is known to have detrimental effects on healthcare staff with regard to both personal and occupational matters. The association between burnout symptoms and work satisfaction in endoscopy nursing staff in Germany has not been studied previously. We aimed to investigate the association between work satisfaction and risk of burnout in endoscopy nursing staff in Germany and to extract predictors for burnout in the area of work satisfaction, which can inform the design of future interventions. Setting: All members of the German Association of Endoscopy Staff in Germany (Deutsche Gesellschaft für Endoskopiefachberufe e.V.—DEGEA) were invited to take part in an online survey. Methods: The total sample consisted of 674 endoscopy staff members. Of those, 579 were female (85.9%) and 95 were male (14.1%). The mean age of the participants was 44.3 years (SD 10.6), with a median age of 46 years, a minimum age of 20, and a maximum age of 64 years. We used confirmatory factor analyses to examine the Maslach burnout inventory (MBI) and, a questionnaire for assessing general and facet-specific job satisfaction (KAFA), regarding their postulated internal structure in our special sample. Canonical correlations were performed to examine the association between work satisfaction and burnout in endoscopy staff members. Results: We were able to replicate the factorial structures of the MBI and the KAFA, both showing an acceptable model fit. The canonical correlation analysis resulted in three canonical functions, with canonical correlations of 0.64 (*p* < 0.001), 0.32 (*p* < 0.001), and 0.17 (*p* < 0.001). The first canonical function revealed that KAFA scales for colleagues, professional development, payment, supervisor, and general job satisfaction were good predictors for less exhaustion, less depersonalization and lack of empathy, and higher personal accomplishment. Commonality analysis revealed that general job satisfaction was the most significant factor in explaining the squared canonical correlation. The second canonical function showed that occupational function and colleagues were good predictors for exhaustion and personal accomplishment. Conclusions: Interventions aimed at ameliorating symptoms of burnout in endoscopy staff should be tailored to address specific needs as experienced by the employees. Therefore, the results of this study could contribute to the design of various interventions, which could be employed to address the issue of work satisfaction and burnout in endoscopy staff most effectively.

## 1. Introduction

Medical and scientific progress and demographic change have made the affordability of solidly financed and highly developed healthcare systems a major challenge for society. In Germany, this development has led to “economization”, which has entered the system as a decisive and much-discussed factor alongside patient orientation [1,2]. Nursing staff in the German healthcare system experience these developments primarily as a high concentration of work [3,4,5]. Various studies show that this can lead to a health risk for the staff, in addition to an increased risk of burnout and a decreased work satisfaction [6,7].

Burnout, as a combination of depersonalization, emotional exhaustion, and reduced personal accomplishment [4], is known to have detrimental effects on healthcare staff with regard to personal and occupational matters [8]. It is important to study burnout in nursing staff because burnout increases staff absenteeism, has negative impacts on the quality of care, increases medical errors, and reduces patient safety [9]. Factors associated with burnout are the hospital service in which nurses work, the tasks performed, the roles they take, and the type of patients treated [10]. In a meta-analysis of data in primary care nursing, 28% of the nursing staff exhibited high emotional exhaustion, high depersonalization was present in 15%, and 31% had a low sense of personal accomplishment [11]. In another study on hospital nurses in Andalusia, 40% presented high levels of burnout [12].

Job stress is closely related to work satisfaction [13]. In turn, the amount of pleasure one feels doing one’s job, known as work satisfaction, is also determined by a number of factors [14,15]. Low work satisfaction with high levels of stress and an unsatisfactory work–life balance can lead to symptoms of burnout [16]. The dual-factor theory of job satisfaction and motivation [17] has been widely applied in nursing research. It posits that there are lower-order extrinsic work needs such as working conditions and salary levels, and higher-order intrinsic needs such as feelings of advancement and value in the nature of the work itself, which lead to higher levels of work satisfaction. In a systematic review, it was found that intrinsic needs such as responsibility and social service contributed to a greater portion of job satisfaction in nurse practitioners (NP), while extrinsic needs such as company policies, working conditions, and compensation were prominent factors in job dissatisfaction [18]. Therefore, administrators should reduce extrinsic sources of job dissatisfaction. Problems with professional growth and collegiality made the greatest contributions to job dissatisfaction. There was a substantial negative correlation between job satisfaction and anticipated turnover in NPs (r = −0.51). Little control over their practice, limited career opportunities, and not being recognized as valuable members of the team were the main reasons for turnover intentions. Dissatisfaction with professional and monetary recognition, lack of administrative support, and lack of collegiality were the main extrinsic factors contributing to job dissatisfaction. The highest job satisfaction for intrinsic factors was observed for time for patient care, quality of care provided, sense of accomplishment, and being challenged by the work performed. Autonomy was found to have the highest association with general job satisfaction [18]. Healthcare providers with higher levels of job satisfaction are more productive, deliver better patient care, and their patients indicate higher levels of satisfaction [19].

As implications for future research, Hoff et al. state in their review of the empirical literature that the potential interrelationships within the same study between job satisfaction, burnout, job stress, and turnover intentions should be studied. Subgroup analyses that compare different groups of nurse practitioners and physician assistants for factors such as age, gender, and employment setting should be performed. [20].

To the best of our knowledge, no studies for the German setting are available to date examining burnout and work satisfaction in endoscopy nursing staff. With this study, we aim to investigate the association between work satisfaction and risk of burnout in endoscopy nursing staff in Germany and to extract predictors for burnout in the area of work satisfaction, which could inform the design of future interventions.

## 2. Materials and Methods

### 2.1. Design and Sample Recruitment

We performed an online quantitative cross-sectional study using the platform Limesurvey. Endoscopy nursing staff were queried about their baseline demographic variables, work satisfaction, and burnout risk. Participation was voluntary, and all the responses were anonymous. 

This survey was performed among endoscopy nursing staff in Germany between January and April 2019. All members of the German Association of Endoscopy Staff in Germany (Deutsche Gesellschaft für Endoskopiefachberufe e.V.—DEGEA) were invited to take part in the study (*n* = approximately 2000). The DEGEA is the largest association of endoscopy staff in Germany, in which the majority of staff holds a membership.

The survey comprised an invitation with a detailed study description, an informed consent form, and the study questionnaire. The short questionnaire to assess general and facet-specific job satisfaction (Kurzfragebogen zur Erfassung von Allgemeiner und Facettenspezifischer Arbeitszufriedenheit—KAFA) and the Maslach burnout inventory (MBI-D) were used in the study. Both instruments have been validated for the German setting. Members of the DEGEA received an email invitation to participate and a link to the study was also published on the DEGEA website. After eight weeks, a reminder to participate in the study was sent to all members. The study was performed in accordance with the Declaration of Helsinki and approved by the data protection commissioner of the University of Siegen. All participants were enrolled in the study according to the principles of informed consent and confidentiality.

### 2.2. Assessment of Work Satisfaction and Risk of Burnout

The questionnaire for assessing general and facet-specific job satisfaction (KAFA) follows the concept of the job descriptive index. Job satisfaction is not directly measured but derived from evaluative descriptions. Positive and negative characteristics and behaviour are specified regarding a superordinate concept (e.g., “My supervisor… is fair” or “is present when needed”). Job satisfaction is regarded as an evaluative judgment of work and aspects of work mirrored in the agreement or non-agreement with positive or negative descriptions. The KAFA measures job satisfaction by “general job satisfaction” and five facets, namely “occupational function”, “colleagues”, “professional development”, “payment”, and “supervisor”, with 5 items each scored on a 5-point scale from “1—totally disagree” to “5—totally agree”. Cronbach’s α ranges from 0.87 to 0.91. Factorial validity could be confirmed by confirmatory factor analyses. The KAFA scales show expected associations with several external criteria. “Payment” correlates with r = 0.36 for monthly gross income. The association between the KAFA scale “occupational function” and job characteristics in line with Hackman and Oldham was r = 0.51. There were negative associations between the KAFA scales and thoughts and intentions to leave (e. g. r = −0.60 between “general job satisfaction” and frequency of leaving thoughts). Furthermore, there were expected associations between incidents at work and facets of work satisfaction. The KAFA scale “colleagues” correlated at r = −0.53 with the frequency of arguments between colleagues and the scale “supervisor” with the frequency of negative (r = −0.43) and positive (r = 0.41) feedback by supervisors [3].

We used the German version of the Maslach burnout inventory (MBI) to assess occupational burnout. The MBI is designed to measure an enduring state of experiencing burnout, an assumption that is borne out by the stability of its scores over time [4]. The MBI is comprised of 22 items, each scored on a 7-point scale from “0—never” to “7—every day”. It consists of 3 subscales: “emotional exhaustion” (9 items), which measures exhaustion at work; depersonalization (5 items), which measures emotional distance to others and loss of empathy; and personal accomplishment (8 items), which measures competence and a positive attitude towards work. The three-factor structure was confirmed; the Cronbach’s α value for the emotional exhaustion scale was 0.85, for the personal accomplishment subscale was 0.71, and for the depersonalization subscale was only 0.48 [5]. Other studies found higher internal consistency for this subscale, with Cronbach alpha values of 0.69, 0.82, and 0.86, respectively [21,22,23]. The convergent and discriminant validity of the MBI was demonstrated. The three-factor structure was also supported in an international study with nurses and in our study with German GPs [22,24].

### 2.3. Statistical Analyses

Missing values were replaced by the k-nearest neighbor algorithm (kNN) using the R package Visualization and Imputation of Missing Values (VIM) [25].

We used Chi-square tests with Cramér V effect size to compare categorical variables and the Welch test with Cohen’s d effect size to compare independent groups [26,27].

We conducted confirmatory factor analysis with the R package lavaan [28] to examine the hypothesized factorial structures of the MBI and the work satisfaction questionnaire in our sample of endoscopy physicians. We used the robust unweighted least squares estimator (ULSMV), as this estimation method makes no distributional assumptions [28,29]. Different model fit statistics were calculated. The χ^2^/df ratio is a badness-of-fit index, as smaller values indicate a better fit [30]. Values between 2 and 5 signal an acceptable model fit [31,32]. The root mean square error of approximation (RMSEA) is a population-based index that relies on the noncentral χ^2^ distribution. It can be regarded as an “error of approximation” index because it assesses the extent to which a model fits reasonably well in the population [33]. Values ≤ 0.08 are considered to indicate an adequate model fit [34]. The standardized root mean square residual (SRMR) was calculated to measure the mean absolute value of covariance residuals [35]. Values below 0.10 indicate a good model fit [36]. The comparative fit index (CFI) and the Tucker–Lewis index (TLI) were not considered, as it was observed that they were sensitive to models with more variables, such as ours [37]. The resulting items and scales were examined by parameters based on classical test theory, such as Cronbach’s α, discriminatory power, or average intercorrelations. Omega coefficients for the applied scales were also computed using the R packages psych and GPArotation, as they have known advantages over Cronbach’s α [38].

We used canonical correlation analysis to examine the association between work satisfaction and burnout in endoscopy nursing staff [39]. According to previous findings, we labelled the 6 scales of the KAFA as independent variables and the 3 scales of the Maslach burnout inventory as dependent variables. The subject/variable ratio was 75:1, and therefore much higher than the recommended 10:1 ratio [40]. We also performed canonical commonality analysis to enhance interpretation of the results [41]. The R packages yacca and yhat were used for calculations.

## 3. Results

### 3.1. Study Sample

The total sample consisted of 674 endoscopy staff members. Of those, 579 were female (85.9 %) and 95 were male (14.1 %). The mean age of the participants was 44.3 years (SD 10.6), with a median age of 46 years, a minimum age of 20, and a maximum age of 64 years.

Participants had worked in the field of endoscopy for 12.8 years on average (SD 8.6). Most of the staff members were specialized endoscopy nurses (*n* = 273, 40.5%), followed by nurses (*n* = 239, 35.5%) and physician assistants (*n* = 188, 27.9%). Numbers add up to more than 100% due to double qualifications. Exactly 509 (75.5 %) were working in a hospital, while 165 (24.5 %) were working in the ambulatory sector, e.g., private practices.

The majority were working full time (*n* = 456, 67.7%), 171 (25.4%) were working more than 20 h per week, 41 (6.1%) were working 10–20 h per week, and 6 (0.9%) were working less than 10 h per week. An executive position was held by 274 (40.7%) staff members.

### 3.2. Methodological Evaluation of the Maslach Burnout Inventory (MBI)

We tested the hypothesized three-factor structure in our sample of endoscopy staff. The confirmatory factor analysis with the robust MLR estimation method still showed an acceptable model fit: χ^2^/df = 5.56, RMSEA = 0.089, SRMR = 0.097.

Two items have a factor loading under the recommended cut-off value of 0.30 [33]. These are items 22, “Blame”, of the depersonalization and lack of empathy factor, with a loading of 0.297; and item 4, “Can understand” of the “personal accomplishment” factor, with a loading of 0.14. All other items have loadings between 0.38 and 0.86.

Intercorrelations of factors are satisfactory, with emotional exhaustion correlating with depersonalization and loss of empathy by r = 0.69 and with personal accomplishment by r = −0.43, while depersonalization and loss of empathy correlates with personal accomplishment by r = −0.42.

The Cronbach’s α coefficient of the emotional exhaustion subscale was 0.88, the omega coefficient was 0.89, and the average inter-item correlation was 0.45. The discriminatory power of the items ranged from 0.47 to 0.79. The Cronbach’s α coefficient of the depersonalization and loss of empathy subscale was 0.66, the omega coefficient was 0.68, and the average inter-item correlation was 0.28. The discriminatory power of the items ranged from 0.24 to 0.56. The Cronbach’s α coefficient of the personal accomplishment subscale was 0.72, the omega coefficient was 0.73, and the average inter-item correlation was 0.24. The discriminatory power of the items ranged from 0.18 (item 4, “Can understand”) to 0.54. The values can be classified as satisfactory to high, except for the reliability of the depersonalization and loss of empathy subscale and the low discriminatory power of item 4.

The mean of the scale emotional exhaustion was 18.9 (SD 11.3), with a median of 18, a minimum of 0, and a maximum of 52. Its distribution deviated significantly from normal distribution: Shapiro–Wilk test, *p* < 0.0001; skewness, *p* < 0.0001 (right-skewed); kurtosis, *p* < 0.0001 (platykurtic). In comparison with a sample of *n* = 2681 German nurses [22], the mean of our endoscopy staff sample was significantly higher (Welch’s *t*-test: t (df = 1117) = 9.89, *p* < 0.001, Cohen’s d = 0.52), and corresponds to a medium effect. The mean of the sample of nurses was 14.4 (SD 8.0).

The mean of the scale depersonalization and loss of empathy was 6.3 (SD 5.5); Huber’s M estimator was 6.0, with a median of 5, a minimum of 0, and a maximum of 29. Its distribution mainly deviated significantly from a normal distribution: Shapiro–Wilk test, *p* < 0.0001; skewness, *p* < 0.0001 (right-skewed); kurtosis, *p* = 0.002 (leptokurtic). In comparison with a sample of *n* = 2681 German nurses [22], the mean of our endoscopy staff sample was significantly lower (Welch’s *t*-test: t (df = 853) = −4.55, *p* < 0.001, Cohen’s d = 0.19), and corresponds to a small effect. The mean of the sample of nurses was 7.4 (SD 6.0). 

The mean of the scale personal accomplishment was 30.1 (SD 8.1), with a median of 30, a minimum of 0, and a maximum of 48. Its distribution mainly deviated from a normal distribution: Shapiro–Wilk test, *p* < 0.0001; skewness, *p* < 0.0001 (left-skewed); kurtosis, *p* = 0.31. In comparison with a sample of *n* = 2681 German nurses [22], the mean of our endoscopy staff sample was significantly lower (Welch’s *t*-test: t (df = 1096) = −19.73, *p* < 0.001, Cohen’s d = 0.82), and corresponds to a large effect. The mean of the sample of nurses was 37.1 (SD 8.6).

In our sample, 177 endoscopy nursing staff members (26.3%) had a high level of emotional exhaustion, 97 (14.4%) showed a high level of depersonalization, and 367 (54.5%) had a low level of personal accomplishment. This has to be regarded with caution, as the cut-off values were derived from the manual of Maslach et al. [4], which are based on a sample from the population of the United States of America. Their comparability to our sample is limited. Cut-off values from appropriate German samples are not available.

### 3.3. Methodological Evaluation of the KAFA

We modified the confirmatory model and tested a six-factor structure, including the general job satisfaction scale, in our endoscopy staff sample. The confirmatory factor analysis with the robust ULSMV estimation method showed a good model fit: χ^2^/df = 1.62, RMSEA = 0.030, SRMR = 0.041, confirming the extended model.

All items have factor loadings over the recommended cut-off value of 0.30 [33]. The range was between 0.37 and 0.90. Intercorrelations between factors were heterogenous. The correlations of all factors with general job satisfaction were all medium, while correlations with factor payment were lower (Appendix A).

The Cronbach’s α coefficient of the occupational function subscale was 0.74; the omega coefficient was 0.76, and the average inter-item correlation was 0.36. The discriminatory power of the items ranged from 0.22 (“uninteresting”) to 0.65. The Cronbach’s α coefficient of the colleagues’ subscale was 0.87; the omega coefficient was 0.88, and the average inter-item correlation was 0.57. The discriminatory power of the items ranged from 0.61 to 0.80. The Cronbach’s α coefficient of the professional development subscale was 0.88; the omega coefficient was 0.88, and the average inter-item correlation was 0.60. The discriminatory power of the items ranged from 0.70 to 0.80. The Cronbach’s α coefficient of the payment subscale was 0.90; the omega coefficient was 0.90, and the average inter-item correlation was 0.63. The discriminatory power of the items ranged from 0.62 to 0.80. The Cronbach’s α coefficient of the supervisor subscale was 0.89, the omega coefficient was 0.89, and the average inter-item correlation was 0.61. The discriminatory power of the items ranged from 0.64 to 0.82. The Cronbach’s α coefficient of the general job satisfaction subscale was 0.80, the omega coefficient was 0.81, and the average inter-item correlation was 0.45. The discriminatory power of the items ranged from 0.44 to 0.69.

The mean of the scale for occupational function was 20.9 (SD 3.4), with a median of 21, a minimum of 9, and a maximum of 25. Its distribution deviated from a normal distribution: Shapiro–Wilk test, *p* < 0.0001; skewness, *p* < 0.0001 (left-skewed); kurtosis, *p* = 0.002 (leptokurtic).

The mean of the scale for colleagues was 19.5 (SD 3.8), with a median of 20, a minimum of 9, and a maximum of 25. Its distribution deviated from a normal distribution: Shapiro–Wilk test, *p* < 0.0001; skewness, *p* < 0.0001 (left-skewed); kurtosis, *p* = 0.004 (platykurtic).

The mean of the scale for professional development was 16.4 (SD 5.2), with a median of 17, a minimum of 5, and a maximum of 25. Its distribution deviated from a normal distribution: Shapiro–Wilk test, *p* < 0.0001; skewness, *p* = 0.006 (left-skewed); kurtosis, *p* < 0.0001 (platykurtic).

The mean of the scale for payment was 15.1 (SD 5.5), with a median of 15, a minimum of 5, and a maximum of 25. Its distribution deviated from a normal distribution: Shapiro–Wilk test, *p* < 0.0001; skewness, *p* = 0.29; kurtosis, *p* < 0.0001 (platykurtic).

The mean of the scale for supervisor was 19.3 (SD 3.6), with a median of 20, a minimum of 5, and a maximum of 25. Its distribution deviated from a normal distribution: Shapiro–Wilk test, *p* < 0.0001; skewness, *p* < 0.0001 (left-skewed); kurtosis, *p* = 0.035 (platykurtic).

The mean of the scale for general job satisfaction was 18.4 (SD 4.7), with a median of 19, a minimum of 5, and a maximum of 25. Its distribution deviated from a normal distribution: Shapiro–Wilk test, *p* < 0.0001; skewness, *p* < 0.0001 (left-skewed); kurtosis, *p* = 0.001 (leptokurtic).

### 3.4. Association between Burnout and Work Satisfaction

The canonical correlation analysis resulted in three canonical functions, with canonical correlations of 0.64 (*p* < 0.001), 0.32 (*p* < 0.001), and 0.17 (*p* < 0.001). The full model across all functions was significant (χ^2^ (18) = 451.45, *p* < 0.001) [41]. All three functions are statistically significant, and the first function accounts for a considerable amount of variance (41.5% versus 10.3% versus 3.0%, respectively), although interpreting squared multiple correlations as indicating the amount of shared variance between two variable sets has been criticized [42,43]. In the following, we interpret only functions 1 and 2, as the canonical correlation of function 3 was low and reached significance just because of the large sample size.

Function 1 revealed that the predictor canonical variate is characterized by colleagues, professional development, payment, supervisor, and general job satisfaction (Table 1). Colleagues displays a pattern of cross-loadings with similar correlations with functions 1, 2, and 3, but a slightly higher value in function 1. The first criterion canonical variate is characterized by exhaustion (r = −0.86), personal accomplishment (r = 0.69), and depersonalization and lack of empathy (r = −0.75) (Table 2). There were also higher cross-loadings of personal accomplishment in functions 2 and 3, and a higher cross-loading of depersonalization and lack of empathy, which also has a high value in function 3. High values in colleagues, professional development, payment, supervisor, and general job satisfaction in particular seem to be good predictors for less exhaustion and less depersonalization and lack of empathy, as both have negative correlations with the first criterion canonical variate (Table 2). Furthermore, they are good predictors for personal accomplishment, as this MBI scale has a positive correlation with the first criterion canonical variate. This means that the more satisfied the endoscopy staff were with colleagues, professional development, payment, supervisor, and general job satisfaction, the less exhausted they felt, the less depersonalization and lack of empathy they experienced, and the higher their personal accomplishment was. The low standardized function coefficients of professional development, payment, and supervisor, and their relatively high correlations with the first canonical variate indicates that the variance of these variables is explained by the other variables. Figure 1 displays the structure correlations (loadings) of the KAFA scales on the first predictor canonical variate and of the structure correlations (loadings) of the MBI scales on the first criterion canonical variate, and visualizes the differential loading patterns and associations between job satisfaction and burnout variables in function 1.

With commonality analysis, it is possible to partition the variance each variable contributes to the explained variance expressed by the squared canonical correlation of each canonical function into unique variance and into common variance, which is variance contributed by a combination of several variables [41]. In function 1, the squared canonical correlation was 0.415 and the general job satisfaction scale contributed 0.069 (16.6% of the squared canonical correlation) to this squared coefficient of the burnout canonical variate. Next was the variance common to colleagues and general job satisfaction, with a coefficient of 0.025 (6.1% of the squared canonical correlation) (Table 3). The burnout canonical variate was mainly explained by the common variance of the KAFA scales. For example, the KAFA supervisor scale explained 13.9% of the variance of the first canonical variate together in combination with other KAFA scales, but 0% unique variance. The work satisfaction canonical variate was primarily explained by variance common to exhaustion and depersonalization and lack of empathy, with a coefficient of 0.125 (30.0% of the squared canonical correlation); variance common to exhaustion, depersonalization and lack of empathy, and personal accomplishment, with a coefficient of 0.08 (19.3% of the squared canonical correlation); and variance unique to exhaustion, with a coefficient of 0.08 (19.3% of the squared canonical correlation).

Function 2 revealed that the predictor canonical variate is characterized by occupational function and colleagues (Appendix A). The relatively high standardized function coefficient for general job satisfaction and its low correlation with the first canonical variate indicates the presence of a suppression effect. The second criterion canonical variate is mainly characterized by exhaustion (r = 0.49) and personal accomplishment (r = 0.50) (Appendix A). High scores for occupational function and colleagues seem to be good predictors for exhaustion and personal accomplishment. This might indicate a pattern of overcommitment. Figure 2 displays the structure correlations (loadings) of the KAFA scales on the second predictor canonical variate and of the structure correlations (loadings) of the MBI scales on the second criterion canonical variate, and visualizes the differential loading patterns and associations between job satisfaction and burnout variables in function 2.

In function 2, the burnout canonical variate with a squared canonical correlation of 0.103 was also explained by several elements. The unique contribution of occupational function with a coefficient of 0.041 (39.4% of the squared canonical correlation) was highest, followed by the unique contribution of general job satisfaction at 0.040 (38.5%) and the unique contribution of colleagues at 0.025 (23.7%). It is apparent that there are suppressor effects in connection with general job satisfaction (Appendix A). Those with higher general job satisfaction and higher satisfaction with colleagues and with their occupational function might be less prone to a pattern of exhaustion and personal accomplishment, which may signal overcommitment. This interpretation might be supported by the negative explained common variance in the second work satisfaction canonical variate, by a combination of exhaustion and depersonalization and lack of empathy with a coefficient of −0.031 (−30.4% of the squared canonical correlation), as this signals a differential association between these variable sets. The second work satisfaction canonical variate is explained mainly by the unique contribution of exhaustion, with a coefficient of 0.0773 (74.9% of the squared canonical correlation), followed by the unique contribution of personal accomplishment, with a coefficient of 0.0383 (37.1% of the squared canonical correlation). These lastly mentioned contributions add up to more than 100%, as there are also negative contributions that sum up to 100%.

The third canonical correlation was low at 0.17. Therefore, the third function should not be interpreted.

## 4. Discussion

Canonical correlation analysis was used to examine the association between work satisfaction and burnout in endoscopy staff. The first canonical function revealed that the KAFA scales for colleagues, professional development, payment, supervisor, and general job satisfaction were good predictors for less exhaustion, less depersonalization and lack of empathy, and higher personal accomplishment. Commonality analysis revealed that general job satisfaction made the greatest contribution to explaining the squared canonical correlation. The second canonical function showed that occupational function and colleagues were good predictors for exhaustion and personal accomplishment. Besides the unique contributions of these KAFA scales and general job satisfaction, there were also suppressor effects showing negative explained variance. Those showing high satisfaction with their occupation and their colleagues might be prone to exhaustion, although they have higher personal accomplishment scores. The suppressor effects might show that those with higher general job satisfaction and higher satisfaction with colleagues and their occupational function might have a lower risk of developing a pattern of exhaustion and personal accomplishment, which may signal overcommitment. The negative explained common variance in the second work satisfaction canonical variate by a combination of exhaustion and depersonalization and lack of empathy supports this argument, as this signals a differential association between these variable sets. Our results corroborate several other findings. Interprofessional teamwork was shown to be a decisive factor for job satisfaction in healthcare [44]. Problems with professional growth and collegiality made the highest contribution towards job dissatisfaction [18]. The prevalence of the burnout dimensions among endoscopy nursing staff in our study were comparable to other studies. Prevalence rates of high emotional exhaustion were almost identical to a Spanish study [10], while the prevalence of high depersonalization was lower (14.4 % versus 30 %) and prevalence of low personal accomplishment was higher (54.5% versus 44%) in our sample. One has to consider that different cut-off values were used and that our sample consisted of a special subgroup of healthcare staff. As there were not high scores in at least two of the burnout subdimensions, one cannot conclude a generally elevated level of burnout in our sample. Our study is in line with the suggestion of Hoff et al. that larger scale surveys by reputable national organizations could achieve the necessary databases for subgroup analyses that compare different groups of nurse practitioners and physician assistants on factors such as age, gender, and employment setting [20].

The investigation of the association between work satisfaction and burnout risk in endoscopy nursing staff revealed interesting results, which have important consequences for clinical management. The results support the hypothesis of burnout being a multidimensional construct, which has to be thoroughly diagnosed. Differential interventions designed for those with specific deficits in certain areas should be delivered. As our results reveal, endoscopy nursing staff members with differential associations between work satisfaction and burnout would need different interventions. The results of our study could contribute substantially to the design of various interventions aimed to ameliorate symptoms of burnout and to increase work satisfaction in endoscopy nursing staff. Future interventions should refer to evidence-based components and tailor them to the specific needs of special staff subgroups. Gómez-Urquiza reported in their meta-analysis that younger nursing staff were more vulnerable to higher emotional exhaustion and depersonalization [45]. Longitudinal studies are needed to document the stability of differential therapeutic interventions in this area. A meta-analysis on coping strategies for burnout revealed that the effects of therapeutic interventions regarding emotional exhaustion and depersonalization could be maintained for one year, while effects regarding personal accomplishment could be maintained for 6 months. A negative correlation was found between personal accomplishment and problem solving [46]. A structural equation model of locus of control and work-related stress predicted 38% of variance in burnout in German nurses. Furthermore, higher work-related stress and burnout were associated with poorer locus of control. Nurses who believe they have less control over events in their lives are supposed to be more vulnerable to stress and burnout. This is important, as other factors such as job satisfaction and commitment are related to locus of control. Social skills training might improve problem solving skills and strengthen the perception of having a higher internal locus of control [47]. Psychological empowerment composed of meaning (fit between job requirements and own ideals and standards), competence (belief in one’s own abilities), self-determination (sense of control over one’s work or autonomy), and impact (ability to influence important work outcomes) is associated with low stress, low burnout, low turnover intentions, high organizational commitment, and high job satisfaction. The meta-analytic correlation between psychological empowerment and job satisfaction was 0.353. Psychological empowerment mediates structural empowerment and job satisfaction. Consequently, psychological empowerment is important in improving the job satisfaction of nurses [6]. In geriatric care workers, emotional exhaustion, depersonalization, negative affectivity, the need to hide negative emotions, and job demands correlate negatively with job satisfaction. There were expected positive associations between depersonalization, emotional exhaustion and negative affectivity, the need to hide negative emotions, and job demands. Correlations were in the medium range. Structural models revealed that negative affectivity predicted burnout and was negatively associated with job satisfaction. Low job status had a strong negative effect on job control. The perceived need to hide negative emotions increased emotional exhaustion and job dissatisfaction, while the perceived need to show positive emotions showed opposite effects. The results support arguments for greater autonomy for the care staff and for the implementation of programs for emotion regulation in healthcare professionals [9]. As can be seen in the empirical literature, the provision of coping strategies, social skills training, psychological empowerment, and emotion regulation could be promising starting points and components of intervention programs for endoscopy nursing staff.

Our study has limitations that should be considered. The survey was sent to members of a professional society, which may not fully represent the population of endoscopy staff members. We collected cross-sectional data based on self-reports, which always have to be interpreted with caution, as such a design does not allow causal inferences between the included variables. We were not able to record longitudinal assessments or to set up an experimental design. The resulting associations in our study should, therefore, be regarded as hypotheses that were generated regarding the design of interventions, which have then to be tested in controlled studies.

## 5. Conclusions

Medium positive correlations were found between job satisfaction and work engagement and organizational citizenship behavior. Increasing the job satisfaction in nurses has the potential to further improve the quality of care and to ensure the provision of a motivated and empowered workforce in an important area for modern society [7]. We could demonstrate differential associations between job satisfaction and burnout in endoscopy nursing staff, showing the multidimensionality of the burnout construct in specific subgroups. This has to be considered as separate norms and tailored empirical studies. Consequently, focusing on these aspects in research and in interventions will contribute to the improved provision of care in the field of gastroenterology. Interventions aimed at ameliorating symptoms of burnout in endoscopy staff should be tailored to address specific needs as experienced by the employees. Therefore, the results of this study could contribute to the design of various interventions that could be employed to address the issue of work satisfaction and burnout in endoscopy staff most effectively.

## Figures and Tables

**Figure 1 ijerph-17-02964-f001:**
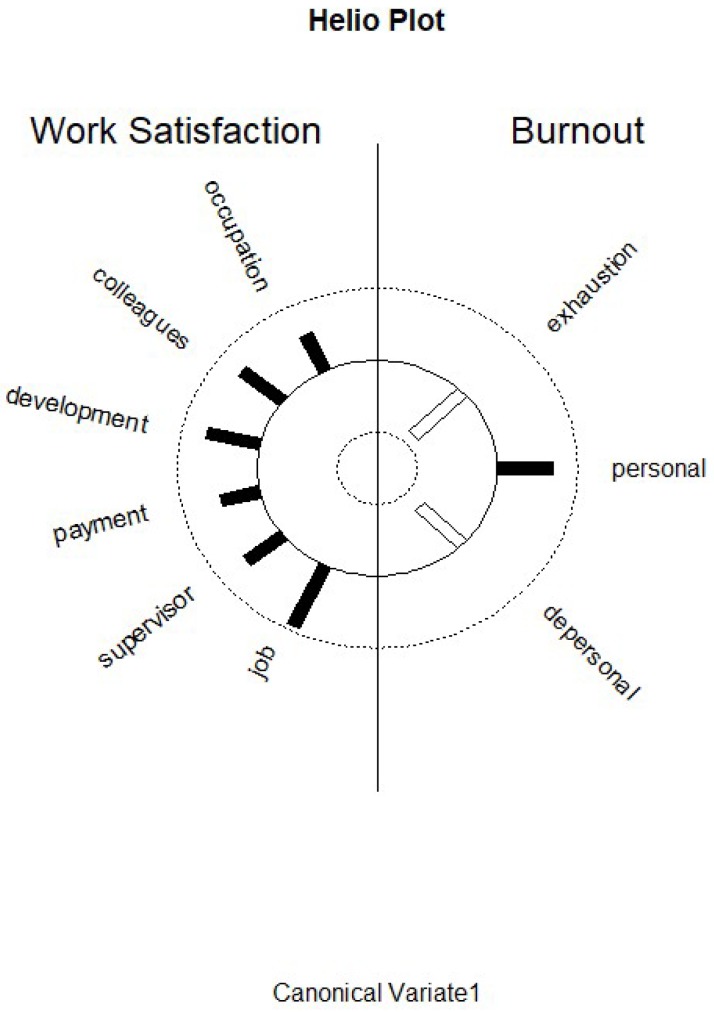
Graphic display of structure correlations (loadings) of the Kurzfragebogen zur Erfassung von Allgemeiner und Facettenspezifischer Arbeitszufriedenheit (KAFA) scales on the first predictor canonical variate and of the structure correlations (loadings) of the Maslach burnout inventory (MBI) scales on the first criterion canonical variate. Black bars correspond to positive correlations and white bars correspond to negative correlations.

**Figure 2 ijerph-17-02964-f002:**
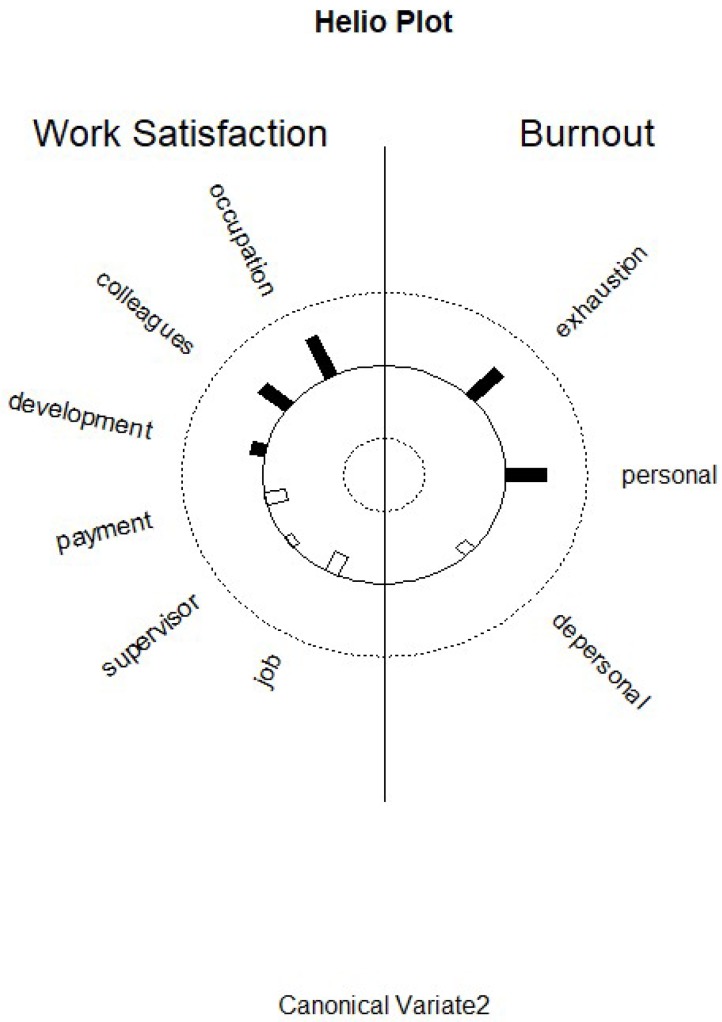
Graphic display of structure correlations (loadings) of the KAFA scales on the second predictor canonical variate and of the structure correlations (loadings) of the MBI scales on the second criterion canonical variate.

**Table 1 ijerph-17-02964-t001:** Standardized canonical coefficients and structure correlations of the first predictor canonical variate.

Predictor Canonical Variate	Standard Canonical Coefficients	Structure Correlations
Occupational function	0.13	0.55
Colleagues	0.25	0.66
Professional development	0.19	0.68
Payment	0.18	0.50
Supervisor	0.01	0.58
General job satisfaction	0.59	0.93

**Table 2 ijerph-17-02964-t002:** Standardized canonical coefficients and structure correlations of the first criterion canonical variate.

Criterion Canonical Variate	Standard Canonical Coefficients	Structure Correlations
Emotional exhaustion	−0.56	−0.86
Personal accomplishment	0.46	0.69
Depersonalization	−0.27	−0.75

**Table 3 ijerph-17-02964-t003:** Main results of commonality analysis for the first prediction canonical variate.

Predictor Canonical Variate	Coefficient	% Total
Unique to general job satisfaction	0.069	16.6
Common to colleagues and general job satisfaction	0.025	6.1
Common to payment and general job satisfaction	0.022	5.3
Common to professional development and general job satisfaction	0.021	5.0

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
