# Peer review of "The Association of Work Satisfaction and Burnout Risk in Endoscopy Nursing Staff—A Cross-Sectional Study Using Canonical Correlation Analysis"

_ijerph, 2020, doi:10.3390/ijerph17082964_

Round 1
Reviewer 1 Report
MATERIALS AND METHODS
Design and Sample recruitment:
- How was the sample chosen? Authors must specify it.
Assessment of work satisfaction and risk of burnout
- Are the MBI and KAFA scales adapted to the German population? Authors must justify their response.
RESULTS
- The results are very extensive. You must eliminate redundant or uninteresting information.
- Also, the authors have cited other research. In "results" there should be no "discussion". Authors should correct this.
DISCUSSION
- The first two paragraphs do not contribute anything new and repeat information about the results.
- The authors do not explain their results well with references to other studies. The authors have not explained their results relative to other studies.
REFERENCES
- Many bibliographies are obsolete and some citations are incomplete. The bibliographic citations used are more than 5 years old (60, 5%). The authors must update and arrange the bibliography.
- There is an updated bibliography of original and meta-analytic articles that should be cited, among Some recent important references are in IJERPH and Journal of Clinical Medicine. Outstanding references that are interesting:
- Cañadas-De la Fuente GA, Vargas C, San Luis C, García I, Cañadas GR, De la Fuente EI. Risk factors and prevalence of burnout syndrome in the nursing profession. Int J Nurs Stud. 2015; 52(1):240-9.
- Gómez-Urquiza JL, Vargas C, De la Fuente EI, Fernández-Castillo R, Cañadas-De la Fuente GA. Age as a Risk Factor for Burnout Syndrome in Nursing Professionals: A Meta-Analytic Study. Res Nurs Health. 2017; 40(2):99-110.
- Cañadas-De la Fuente GA, Ortega E, Ramirez-Baena L, De la Fuente-Solana EI, Vargas C, Gómez-Urquiza JL. Gender, Marital Status, and Children as Risk Factors for Burnout in Nurses: A Meta-Analytic Study. Int J Environ Res Public Health. 2018; 15(10). pii: E2102.
- Ramirez-Baena L, Ortega-Campos E, Gómez-Urquiza JL, Cañadas-De la Fuente GR, De la Fuente-Solana EI, Cañadas-De la Fuente GA. A multicentre study of burnout prevalence and related psychological variables in medical area hospital nurses. J Clin Med. 2019; 8(1). pii: E92.
Author Response
Reviewer 1
MATERIALS AND METHODS
Design and Sample recruitment:
- How was the sample chosen? Authors must specify it.
Response:
More details of the organization “DEGEA” to clarify the choice of sample was now added to the text:
“This survey was performed among endoscopy nursing staff in Germany between January and April 2019. All members of the German Association of Endoscopy Staff in Germany (DEGEA – Deutsche Gesellschaft für Endoskopiefachberufe e.V.) were invited to take part in the study (n = approx. 2000). The DEGEA is the largest association of endoscopy staff in Germany in which the majority of staff holds a membership. “
Assessment of work satisfaction and risk of burnout
- Are the MBI and KAFA scales adapted to the German population? Authors must justify their response.
Response:
Yes, both instruments have already been adapted to the German population and also validated. Respective literature was cited. This has now been added to the manuscript:
“Both instruments have been validated for the German setting.”
RESULTS
- The results are very extensive. You must eliminate redundant or uninteresting information.
Response:
Please excuse us but the other reviewers expressed different opinions. Another reviewer even urged us to perform further analyses. Nevertheless, we went through our Results section and compared it to our successful submissions to other high-ranked journals.
We gave a short description of the study sample.
We examined the postulated internal structure of the MBI by confirmatory factor analysis and reported this. This is important to verify in such a specific subgroup, also internal consistency. Descriptive measures should always be reported and the comparison to a sample of German nurses is a reference to a similar group. The same holds for the other instrument, the KAFA.
The canonical correlation analyses are described according to the publication of Nimon (2010) on which our RCode is based. Shortening these explanations would distort relevant conclusions. Please excuse us, we are aware that the results are extensive, but these are based on statistical methods which are necessary to build up precise results in this specific subgroup.
- Also, the authors have cited other research. In "results" there should be no "discussion". Authors should correct this.
Response:
We compared the MBI results of our sample to results of a sample of German nurses and labeled the effects sizes as small, medium, or high. These are statistical comparisons and not Discussion in our opinion. A statistical discussion regarding the redundancy index was deleted from the Results section according to the reviewer’s suggestion.
DISCUSSION
- The first two paragraphs do not contribute anything new and repeat information about the results.
Response:
The paragraphs were deleted from the manuscript.
- The authors do not explain their results well with references to other studies. The authors have not explained their results relative to other studies.
Response:
The results of our study were more directly set in connection to other studies with reference to the proposed literature.
REFERENCES
- Many bibliographies are obsolete and some citations are incomplete. The bibliographic citations used are more than 5 years old (60, 5%). The authors must update and arrange the bibliography.
- There is an updated bibliography of original and meta-analytic articles that should be cited, among Some recent important references are in IJERPH and Journal of Clinical Medicine. Outstanding references that are interesting:
- Cañadas-De la Fuente GA, Vargas C, San Luis C, García I, Cañadas GR, De la Fuente EI. Risk factors and prevalence of burnout syndrome in the nursing profession. Int J Nurs Stud. 2015; 52(1):240-9.
- Gómez-Urquiza JL, Vargas C, De la Fuente EI, Fernández-Castillo R, Cañadas-De la Fuente GA. Age as a Risk Factor for Burnout Syndrome in Nursing Professionals: A Meta-Analytic Study. Res Nurs Health. 2017; 40(2):99-110.
- Cañadas-De la Fuente GA, Ortega E, Ramirez-Baena L, De la Fuente-Solana EI, Vargas C, Gómez-Urquiza JL. Gender, Marital Status, and Children as Risk Factors for Burnout in Nurses: A Meta-Analytic Study. Int J Environ Res Public Health. 2018; 15(10). pii: E2102.
- Ramirez-Baena L, Ortega-Campos E, Gómez-Urquiza JL, Cañadas-De la Fuente GR, De la Fuente-Solana EI, Cañadas-De la Fuente GA. A multicentre study of burnout prevalence and related psychological variables in medical area hospital nurses. J Clin Med. 2019; 8(1). pii: E92.
Response:
A lot of the literature cited is associated with methodological issues and therefore often older than 5 years. We updated the references and also thank you for the proposed literature, which was cited and added to the manuscript.
Reviewer 2 Report
Abstract.
It is not necessary to put any number before each section. For example: (1).
The Background should not include a design part [20-22].
The Method should start by saying the type of study performed. Study results have been included in the Method.
The conclusions do not respond to the main objective:
Aim: to investigate the association between work satisfaction and risk of burnout
Conclusions related to a possible intervention
Keywords.
Risk of burnout is not a MeSH term, Burnout yes.
Introduction
There is information that should be included in the Discussion [89-90]
Materials and Methods
Very good approach to statistical analysis
Rest
The images look cloudy
It's correct.
Author Response
Reviewer 2
Overall comment: Thank you for valuable comments on our manuscript.
Abstract.
It is not necessary to put any number before each section. For example: (1).
Response: The numbers have been deleted.
The Background should not include a design part [20-22].
Response: The design part was deleted.
The Method should start by saying the type of study performed. Study results have been included in the Method.
Response:
At the beginning of the Materials and Methods section we stated “We performed an online quantitative cross-sectional survey study using the platform Limesurvey.” We did not include results of our current study in the Materials and Methods section. It is common to demonstrate empirical evidence from previous studies regarding the reliability and the validity of the instruments used in the respective study. Therefore, we referred to previous studies to demonstrate the reliability and validity of the instruments used in our current study.
The conclusions do not respond to the main objective:
Aim: to investigate the association between work satisfaction and risk of burnout
Conclusions related to a possible intervention
Response: The conclusions regarding our main objective were expanded.
Keywords.
Risk of burnout is not a MeSH term, Burnout yes
Response: Thank you, this has been corrected.
Introduction
There is information that should be included in the Discussion [89-90]
Response: This statement was transferred to the discussion.
Materials and Methods
Very good approach to statistical analysis
Response: We thank the reviewer for this positive judgement of the methodology chosen.
Rest
The images look cloudy
Response: All images have been uploaded separately in higher resolution.
Reviewer 3 Report
The topic of the research is of interest to know the health status of health professionals, in this case of endoscopy nurses in Germany. Furthermore, there is little evidence on this line among nurses in Germany.
Comments
Abstract: In the results, the authors explain: .... The first canonical function revealed that KAFA scales for colleagues, professional development, payment, supervisor, and general job satisfaction were good predictors for less exhaustion, less depersonalization / lack of empathy and higher personal accomplishment. "good predictors ..." in a cross-sectional design study is very confusing.
Introduction: It is incomplete, the authors explain some evidences without scientific references or there are doubts if all the information belongs to the same reference. The information pending to be cited or confirmed is detailed below:
- Various studies show that this can lead to a health risk for the staff alongside an increased burnout risk and a decrease in work satisfaction… What studies??
- Burnout as a combination of depersonalization, emotional exhaustion, and reduced personal accomplishment. Reference?
- It’s important to study burnout in nursing staff because burnout increases staff absenteeism, has negative impact on the quality of care, increases medical errors, and reduces patient safety. Reference?
- Factors associated with burnout are the hospital service in which nurses work, the tasks performed, the roles they take, and the type of patients treated. Reference?
- Therefore, administrators should reduce extrinsic sources of job dissatisfaction. Problems with professional growth and collegiality made the highest contribution to job dissatisfaction. There was a substantial negative correlation between job satisfaction and anticipated turnover in NPs (r=-.51). Little control over their practice, limited career opportunities, and not being recognized as a valuable member of the team were the main reasons for turnover intentions. Dissatisfaction with professional and monetary recognition, lack of administrative support, and lack of collegiality were the main extrinsic factors contributing to job dissatisfaction. The highest job satisfaction for intrinsic factors was observed for time for patient care, quality of care provided, sense of accomplishment, being challenged by the work performed. Autonomy was found to have 80 the highest association with general job satisfaction [13]. All this information comes from this reference?
Aims: ……and to extract predictors for burnout in the area of work satisfaction which can inform the design of future interventions… Due to the study design, only associations between variables can be indicated
Materials and Methods
The authors analysed the association between job satisfaction and burnout using canonical correlation analysis. The analysis carried out is correct because It’s useful when there are several possible dependent variables, but with longitudinal, cohort designs, etc. It would be interesting to comment on it in the limitations.
Results:
Can the authors clarify what in practice means? …while 165 (24.5 %) were working in practice.
In relation to the burnout values of the nurses, the mean scores obtained indicate low levels in each subdimension. Based on the manual of Maslach et al. participants are assigned to a high degree of burnout with a sum score ≥ 27 in the subscale EE, ≥ 13 in DP and ≤ 31 in PA (Maslach C, Jackson SE, Leiter MP. Maslach burnout inventory manual: third edition. Palo Alto, California: Consulting Psychologists Press, Inc .; 1996). It would be interesting to know the percentage of nurses with high levels of DP and EE and low levels of PA, ie at risk of burnout.
Discussion
The results do not allow to affirm:
Structural models revealed that negative affectivity predicted burnout ……
Author Response
Reviewer 3:
The topic of the research is of interest to know the health status of health professionals, in this case of endoscopy nurses in Germany. Furthermore, there is little evidence on this line among nurses in Germany.
Comments
Abstract: In the results, the authors explain: .... The first canonical function revealed that KAFA scales for colleagues, professional development, payment, supervisor, and general job satisfaction were good predictors for less exhaustion, less depersonalization / lack of empathy and higher personal accomplishment. "good predictors ..." in a cross-sectional design study is very confusing.
Response:
We thank the reviewer for valuable comments on our manuscript.
This term might indeed seem confusing in this context but this is a common technical term in statistical procedures based on regression. It does not necessarily mean that you predict an event in the future. Jenkins & Gavreau state that in cross-sectional studies, data about predictor variables and outcomes for individual study subjects are collected at the same point in time. This type of study does not involve any follow-up, nevertheless it can be more efficient than a cohort study for many types of questions (Jenkins & Gavreau, 2006, https://doi.org/10.1016/B978-012133570-0/50016-0). Nimon (2010) also gives an example of a cross-sectional study with a predictor and a criterion set of variables both recorded at the same time which illustrates that predictor is a common statistical term independent of study design.
Introduction: It is incomplete, the authors explain some evidences without scientific references or there are doubts if all the information belongs to the same reference. The information pending to be cited or confirmed is detailed below:
- Various studies show that this can lead to a health risk for the staff alongside an increased burnout risk and a decrease in work satisfaction… What studies??
- Burnout as a combination of depersonalization, emotional exhaustion, and reduced personal accomplishment. Reference? Maslach et al
- It’s important to study burnout in nursing staff because burnout increases staff absenteeism, has negative impact on the quality of care, increases medical errors, and reduces patient safety. Reference?
- Factors associated with burnout are the hospital service in which nurses work, the tasks performed, the roles they take, and the type of patients treated. Reference?
- Therefore, administrators should reduce extrinsic sources of job dissatisfaction. Problems with professional growth and collegiality made the highest contribution to job dissatisfaction. There was a substantial negative correlation between job satisfaction and anticipated turnover in NPs (r=-.51). Little control over their practice, limited career opportunities, and not being recognized as a valuable member of the team were the main reasons for turnover intentions. Dissatisfaction with professional and monetary recognition, lack of administrative support, and lack of collegiality were the main extrinsic factors contributing to job dissatisfaction. The highest job satisfaction for intrinsic factors was observed for time for patient care, quality of care provided, sense of accomplishment, being challenged by the work performed. Autonomy was found to have 80 the highest association with general job satisfaction [13]. All this information originates from this reference
Response:
- Studies of Lu et al. (2019) and Li et al. (2018) were added. 2. Reference to Maslach et al. was added. 3. Study of Rouxel et al. (2016) was added. 4. Study of La Canadas-De la Fuente (2015) was added. 5. All cited information originates from Han et al. (2018) as this is an important systematic review.
Aims: ……and to extract predictors for burnout in the area of work satisfaction which can inform the design of future interventions… Due to the study design, only associations between variables can be indicated
Response: We added a respective statement in the limitations section.
Materials and Methods
The authors analysed the association between job satisfaction and burnout using canonical correlation analysis. The analysis carried out is correct because It’s useful when there are several possible dependent variables, but with longitudinal, cohort designs, etc. It would be interesting to comment on it in the limitations.
Response: We added a respective statement in the limitations section.
Results:
Can the authors clarify what in practice means? …while 165 (24.5 %) were working in practice.
Response:
“Practice” refers to the ambulatory sector. This has now been added to the manuscript: “…while 165 (24.5 %) were working in the ambulatory sector, e.g. private practices.”
In relation to the burnout values of the nurses, the mean scores obtained indicate low levels in each subdimension. Based on the manual of Maslach et al. participants are assigned to a high degree of burnout with a sum score ≥ 27 in the subscale EE, ≥ 13 in DP and ≤ 31 in PA (Maslach C, Jackson SE, Leiter MP. Maslach burnout inventory manual: third edition. Palo Alto, California: Consulting Psychologists Press, Inc .; 1996). It would be interesting to know the percentage of nurses with high levels of DP and EE and low levels of PA, ie at risk of burnout.
Response:
In our sample, 177 endoscopy nursing staff members (26.3%) had a high level of emotional exhaustion, 97 (14.4%) showed a high level of depersonalisation, and 367 (54.5%) had a low level of personal accomplishment. This has to be regarded with caution as the cut-off values were derived from the manual of Maslach et al. which are based on a sample from the population of the United States of America. Their comparability to our sample is limited. Cut-off values from appropriate German samples are not available.
Discussion
The results do not allow to affirm:
Structural models revealed that negative affectivity predicted burnout ……
Response:
This statement referred to the empirical literature, not to the results of our study. This is a result of the study of Rouxel et al., 2016). We included/corrected the respective citation to make this more clear.
Round 2
Reviewer 1 Report
Dear authors,
The changes are satisfactory. Congratulations on your work.
Best regards
Author Response
Thank you very much!
Reviewer 2 Report
All comments have been resolved. Congratulations on your article
Author Response
Thank you very much!
Reviewer 3 Report
I agree with some of the changes made by the authors.
Comments: Regarding their results, you explain: .....In our sample, endoscopy nursing staff members (26.3%) had a high level of emotional exhaustion, 97 (14.4%) showed a high level of depersonalization, and (54.5%) had a low level of personal accomplishment. This has to be regarded with caution as the cut-off values were derived from the manual of Maslach et al. [4] which are based on a sample from the population of the United States of America. Their comparability to our sample is limited. Cut-off values from appropriate German samples are not available....
Based on the results presented, the sample of participants presents only unfavorable percentages in the dimension of personal accomplishement, in the other two dimensions there is no more than 33.3% of the sample in the high expected level, so it cannot be stated that there are high levels neither emotional exhaustion nor depersonalization. To have burnout (statistically) means having at least high scores in two of the three subdimensions. This is one of the reasons why the burnout syndrome in some studies has been oversized.
I suggest reviewing the comments on the results obtained in the burnout dimensions. Some of the studies that authors compare have also oversized burnout syndrome in their results.
Author Response
We thank the reviewer for the comment which we understand. The paper was adjusted accordingly: "
"The prevalence of the burnout dimensions among endoscopy nursing staff in our study were comparable to other studies. Prevalence rates of high emotional exhaustion were almost identical to a Spanish study [10] while prevalence of high depersonalisation was lower (14.4 % versus 30 %), and prevalence of low personal accomplishment was higher (54.5 % versus 44 %) in our sample. One has to consider that different cut-off values were used and that our sample consisted of a special subgroup of healthcare staff. As there were not high scores in at least two of the burnout subdimensions, one cannot conclude a generally elevated level of burnout in our sample.Our study is in line with the suggestion of Hoff et al. that larger scale surveys by reputable national organizations could achieve the necessary databases for subgroup analyses that compare different groups of nurse practitioners and physician assistants on factors like age, gender, and employment setting [20]."
We hope that the reviewer is fine with this adjustment.